# Role of Abscisic Acid in the Whole-Body Regulation of Glucose Uptake and Metabolism

**DOI:** 10.3390/nu17010013

**Published:** 2024-12-24

**Authors:** Sonia Spinelli, Zelle Humma, Mirko Magnone, Elena Zocchi, Laura Sturla

**Affiliations:** 1Laboratory of Molecular Nephrology, IRCCS Istituto Giannina Gaslini, Via Gerolamo Gaslini 5, 16147 Genova, Italy; soniaspinelli@gaslini.org; 2Department of Experimental Medicine, Section of Biochemistry, University of Genova, Viale Benedetto XV 1, 16132 Genova, Italy; hummabiochem@gmail.com (Z.H.); mirko.magnone@unige.it (M.M.); ezocchi@unige.it (E.Z.)

**Keywords:** abscisic acid, insulin independent mechanism, metabolic syndrome, diabetes, LANCL1, LANCL2

## Abstract

Abscisic acid (ABA) is a hormone with a long evolutionary history, dating back to the earliest living organisms, of which modern (ABA-producing) cyanobacteria are likely descendants, which existed long before the separation of the plant and animal kingdoms, with a conserved role as signals regulating cell responses to environmental challenges. In mammals, along with the anti-inflammatory and neuroprotective function of ABA, nanomolar ABA regulates the metabolic response to glucose availability by stimulating glucose uptake in skeletal muscle and adipose tissue via an insulin-independent mechanism and increasing metabolic energy production and also dissipation in brown and white adipocytes. Chronic ABA intake of micrograms per Kg body weight improves blood glucose, lipids, and morphometric parameters (waist circumference and body mass index) in borderline subjects for prediabetes and metabolic syndrome. This review summarizes the most recent in vitro and in vivo data obtained with nanomolar ABA, the involvement of the receptors LANCL1 and LANCL2 in the hormone’s action, and the importance of mammals’ endowment with two distinct hormones governing the metabolic response to glucose availability. Finally, unresolved issues and future directions for the clinical use of ABA in diabetes are discussed.

## 1. Introduction

Abscisic acid (ABA) is a terpenoid plant hormone (Figure 1) that regulates many physiological processes in plants, acting as a “stress” signal and triggering plant responses to different environmental stimuli, including light, water, and nutrient availability [1,2,3]. While ABA is commonly linked with plants, more recent research has uncovered a role for this hormone also in several physiological processes in animals, including mammals [4,5,6,7]. Glycemic control is the homeostatic process of keeping blood glucose levels constant in the face of discontinuous nutrient availability, to maintain a steady glucose supply to tissues, particularly neurons. Released by beta-pancreatic cells stimulated by high glucose levels, insulin is the major hormone in humans that stimulates tissue glucose uptake and metabolism, thus reducing glycemia and converting glucose into storage forms, such as glycogen and lipids (fatty acids, triglycerides, and VLDL), for future use by the organism under conditions of low nutrient availability [8,9,10]. New research suggests that ABA also can influence glucose metabolism and glycemic control in mammals. Several studies have shown that ABA stimulates myocyte and adipocyte glucose uptake in an insulin-independent manner and promotes oxidative glucose metabolism in muscle and adipose tissue, leading to metabolic energy production as well as its dissipation as heat [11,12]. This review aims to provide a comprehensive summary of the most recent findings in this field, highlighting the significant advancements and key developments that have emerged. Additionally, it will explore potential future directions for research, identifying areas that require further investigation and could lead to new insights.

### 1.1. Role of ABA in Plants

ABA is a hormone first described and extensively investigated in plants. ABA is involved in several plant developmental processes, such as seed and bud maturation, seed dormancy and germination, fruit development, and ripening. It also plays a key regulatory role in the response of plants to environmental challenges such as drought, soil salinity, cold tolerance, heat stress, and heavy metal ion tolerance.

As an example of its function in drought tolerance, ABA is produced in the roots in response to decreased soil water potential. ABA then acts on the leaves, rapidly altering the osmotic potential of stomatal guard cells, which causes them to shrink and the stomata to close. During times of water scarcity, the stomatal closure induced by ABA minimizes water evaporation, helping to conserve moisture in the leaves and prevent additional water loss. For an update on the extensive literature covering the key role of ABA in the control of developmental processes and plant response to abiotic stress, the reader is redirected to recent and extensive reviews [14,15].

### 1.2. ABA Content in Fruits and Vegetables

ABA is present in a variety of fruits and vegetables, some of which are shown in Table 1. Although most of these plant sources of ABA contain too low amounts of the hormone to be eligible as exogenous ABA sources to humans, it is possible to obtain ABA-enriched fruit extracts, which are indeed already commercially available as food supplements.

### 1.3. Manifold Roles of ABA as an Endogenous Animal Hormone

More than 20 years after the discovery of ABA in plants (the first report was published in 1963) [17,18,19], ABA was also described in Metazoa, ranging from sponges to mammals, including humans. In 1986, Le Page-Degivry et al. first reported that ABA was present in the brain of mice that were fed a synthetic diet devoid of ABA in much higher amounts than in animals fed a regular (ABA-containing) chow, supporting the endogenous origin of ABA [20]. Fifteen years later, studies on early Metazoa (sponges and hydroids) indicated a cross-kingdom conserved role for ABA as an endogenous stress hormone [21,22,23] and paved the way for studies on more complex animals. In 2012, the finding that plasma ABA levels rise after a glucose intake in healthy individuals, reaching concentrations in the low nanomolar range, offered strong evidence that ABA is an endogenous hormone in animals [24]. Finally, the identification of the LANCL proteins as mammalian ABA receptors, LANCL2 in 2011 and LANCL1 in 2021, provided valuable information on ABA functions and its signaling pathway in overexpressing vs. silenced cells and in KO mice (Figure 2) [25,26,27,28,29].

Here, we only touch upon two current areas of investigation regarding the non-metabolic effects of ABA in mammalian physiology, i.e., inflammation (arguably, THE stress response by definition) and neuroprotection.

In rodent models of inflammatory bowel disease, atherosclerosis, and viral infection, exogenous ABA has been found to exert anti-inflammatory effects [30,31,32,33,34,35,36,37]. However, ABA is also autocrinally produced by several innate immune cell types (granulocytes, monocyte-macrophages, and microglia) challenged with pro-inflammatory stimuli and exogenous ABA stimulates several pro-inflammatory activities of these cell types [13,38,39,40]. Thus, the roles of ABA in inflammation and the concentrations exerting different effects require further investigation.

A cardio- and neuroprotective effect of ABA is emerging from recent studies on cardiomyocytes and neuronal cells. In cardiomyocytes, the ABA/LANCL hormone/receptor system regulates the activation of a complex transcriptional mechanism, which results in the stimulation of not only metabolic energy production and contractile and ion channel protein transcription but also the activation of antioxidant defense mechanisms, reducing cell damage after hypoxia/reoxygenation [41,42,43,44]. In neuronal cells, the neuroprotective effects of ABA in various animal and cell models of neurodegenerative diseases seem to be attributed to its trophic, anti-inflammatory, and antioxidant properties [45,46,47,48,49,50,51,52,53,54].

While studies on the cardio- and neuroprotective effects of ABA have not yet reached the clinical setting, a consensus emerges from recent independent studies regarding the beneficial effect of nutraceutical ABA on glycemia control [12,16,55,56]. This organismic effect of ABA is the result of the activation in muscle and adipose cells, which together represent the bulk of body weight and are capable of significantly affecting whole-body glucose metabolism and several metabolic responses, namely increased glucose uptake and oxidative metabolism, elevated mitochondrial mass and ATP content, and steeper proton gradient (ΔΨ), in the face of its increased dissipation [29,57].

### 1.4. ABA Receptors and Signaling Pathways

The conservation of ABA across both plant and animal kingdoms has led to investigations into whether its receptors might also be conserved. While the current consensus identifies the PYR/PYL/RCAR proteins as the main constituents of the ABA perception cascade in plants [58,59,60], one of the earliest proteins proposed as an ABA receptor in Arabidopsis was the membrane-bound, G protein-coupled receptor GCR2 [61,62], which exhibits a significant similarity in amino acid sequence with the lanthionine synthetase C-like (LANCL) protein family of mammals [41]. The LANCL protein family shares structural features with the prokaryotic lanthionine synthetase C proteins, which play a role in producing lantibiotics like nisin. These lantibiotics are commonly used to inhibit bacterial growth in food products [63]. However, because lantibiotics are not generated in animals, mammalian LANCL proteins must operate differently from bacterial LanC protein [64]. Indeed, compelling evidence that mammalian LANCL proteins are not involved in lanthionine synthesis was provided by studies on triple KO mice [65]. The human genome contains three LANCL genes, LANCL1, LANCL2, and LANCL3, which are located on chromosomes 2, 7, and X, respectively [66,67,68]. While LANCL1 and LANCL2 are present in almost every tissue, particularly in the brain (see https://www.proteinatlas.org/search/LANCL, accessed on 20 October 2024), both as transcripts and at the protein level, the transcription of LANCL3 is very low, suggesting that it may be a pseudogene [69,70,71].

LANCL2 was required for ABA binding and signaling in four different cell types from two mammalian species (human granulocytes and HeLa cells, rat microglia and insulinoma cells): (i) LANCL2 silencing prevented the ABA-induced increase in cytosolic free calcium [Ca^2+^]_cyt_ and cAMP in human granulocytes and also prevented the ABA-triggered functional response in these cells [25]; (ii) LANCL2 overexpression conversely enhanced the ABA-induced [Ca^2+^]_cyt_ increase in granulocytes and enabled ABA responsiveness in CD38^+^ HeLa cells, as indicated by elevated levels of [Ca^2+^]cyt and cAMP; and finally, in two separate rat insulinoma cell lines, silencing LANCL2 abolished the ABA-induced biochemical (increase in [Ca^2+^]_cyt_ and cAMP) and functional (insulin release) responses. In vitro binding assays with recombinant LANCL2 and LANCL1 and gene silencing experiments on cells revealed direct ABA binding to both proteins and their role in mediating the cell functional responses to the hormone [25,29,72,73].

The ABA signaling pathway in mammalian cells has been explored in some depth in human granulocytes and rat cardiomyocytes.

In granulocytes, LANCL2 activates both adenylate cyclase (AC) and phospholipase C, resulting in an increase in [Ca^2+^]_cyt_ mediated both by IP3 and by CD38-produced cADPR [13]. The conservation of this ABA signaling pathway from sponges and hydroids to human innate immune cells, involving adenylate cyclase, cAMP, the protein kinase A (PKA)-dependent activation of the ADP-ribosyl cyclase CD38, cADPR overproduction, and [Ca^2+^]_cyt_ rise, indicates its early evolution as a mechanism to enable adaptation to environmental challenges by eliciting an intracellular Ca^2+^ wave, arguably the most ancient and conserved trigger of cell functional responses.

In rat cardiomyocytes and human adipocytes, cell types rather more specialized than innate immune cells, a role for AMPK and the orphan receptor/transcription factor ERRα in mediating ABA signaling downstream of LANCL2 has been recently reported. In these cell types, ABA activates a complex transcriptional program involving mitochondrial biogenesis, oxidative metabolism, oxphos uncoupling, and antioxidant protection, via the activation of AMPK and ERRα [42,57] (Figure 3).

## 2. Role of ABA in the Control of Glycemia

### 2.1. Plasma ABA in Healthy and Diabetic Subjects

In healthy subjects, a glucose load leads to a fivefold physiological increase in plasma ABA (pABA), providing compelling evidence that identifies ABA as an endogenous hormone released/produced in response to an increase in glycemia [24]. In mammals, two additional key peptide hormones are released after a glucose load: pancreatic islet β-cells release insulin, while intestinal endocrine cells release glucagon-like peptide 1 (GLP-1). Insulin is crucial for promoting the uptake of glucose by (insulin-dependent) cells, thereby reducing blood glucose levels. GLP-1, on the other hand, has multiple roles; it not only stimulates the release of insulin in response to elevated blood glucose levels but also inhibits the secretion of glucagon by pancreatic α-cells [75,76]. Interestingly, ABA triggers a glucose-independent release of GLP-1 in the human L cell line hNCI-H716 via a cAMP/PKA-dependent pathway. This mechanism results in GLP-1 secretion from the proximal part of the small intestine and insulin release from the pancreas. In turn, GLP-1 promotes ABA secretion from the pancreas in a biphasic pattern. Notably, a positive correlation was observed between the area under the curve (AUC) for ABA and the AUC for insulin following GLP-1 administration. These findings suggest the presence of a bidirectional interaction between GLP-1 and ABA, where ABA stimulates GLP-1 release, and GLP-1 reciprocally enhances ABA secretion [77,78].

The rise in pABA that normally occurs after an oral glucose load was found to be reduced in type 2 diabetes (T2D) patients compared to healthy controls [79]. A similar pattern was observed in women with gestational diabetes (GDM) compared to pregnant controls. However, at 8–12 weeks postpartum, both fasting pABA levels and the pABA response to glucose were normalized in the GDM subjects, along with their glucose tolerance. Fasting pABA levels were assessed before and after biliopancreatic diversion (BPD) in both obese non-diabetic individuals and obese patients with T2D, the latter group showing the resolution of diabetes post-surgery. One month after BPD, basal pABA levels were significantly higher in both T2D patients and healthy individuals, accompanied by a decrease in fasting plasma glucose compared to pre-BPD levels [79]. These results suggest a critical role for pABA in maintaining normal glucose tolerance and point to a potential ABA-dependent mechanism underlying the diabetic condition. Indeed, the discovery of a second hormone capable of boosting muscle glucose absorption in addition to insulin would have substantial implications in diabetes mellitus, where insulin insufficiency or resistance reduces glucose tolerance.

The fasting pABA levels observed in T2D patients were significantly higher (twofold) relative to controls, suggesting a condition of resistance to the effect of ABA, as occurs with insulin. Moreover, the distribution of the ABAp values was normal in controls but not in T2D subjects [79], possibly due to the heterogeneity of ABA-related dysfunctions occurring in T2D.

In contrast, pABA was undetectable or extremely low in T1D patients, suggesting that β-cells could be the primary source of endogenous ABA in humans. Consequently, the destruction of β-cells in T1D significantly decreases the availability of both hormones that regulate blood glucose levels, only one of which is currently replaced by therapy [79]. Because insulin and ABA have unique metabolic activities, supplementing with one hormone will not restore the function of the other.

The effect of ABA in a murine model of streptozotocin-induced T1D was recently studied [80]. Diabetic mice given a single oral dose of ABA alongside a low-dose subcutaneous insulin treatment exhibited a notably improved glycemic profile compared to controls receiving insulin alone. After four weeks of ABA treatment, the response to a glucose load in diabetic mice was significantly enhanced, with a marked improvement in glycemic control. Additionally, there was an increase in the expression of both the insulin receptor and glycolytic enzymes in muscle tissue. Additionally, skeletal muscle from diabetic mice treated with ABA showed significantly higher transcription and protein expression of AMPK, PGC1-α, and GLUT4 compared to the untreated controls. Supplementing insulin with ABA shows promise in reducing the required insulin dose in T1D, thereby lowering the risk of hypoglycemia and enhancing muscle insulin sensitivity and glucose consumption [80].

### 2.2. ABA Ameliorates Glucose Tolerance in Mice and Healthy Humans

These observations prompted an in vivo investigation into whether ABA’s impact on tissue glucose uptake and glucose tolerance could occur independently of insulin. In a rat study, animals were given an oral glucose load alone (control), with synthetic ABA (1 µg/kg body weight), or with an aqueous fruit extract providing the same ABA dose. Both the extract-treated and synthetic ABA-treated groups exhibited significantly lower area under the curve (AUC) values for glycemia and insulinemia compared to the control group, indicating that a fruit extract titrated in ABA recapitulated the effect of pure ABA, identifying ABA as the active component responsible for these metabolic effects [16]. Notably, when the animals underwent an OGTT with synthetic ABA at a dose of 100 mg/Kg, the AUC of insulinemia remained unchanged compared to the control group, suggesting that at this high dose, ABA did not inhibit insulin release, even though it lowered blood glucose levels. In a separate study with human volunteers, a fruit extract providing about 0.5 µg/kg body weight (BW) was tested in conjunction with carbohydrate-rich meals, such as breakfast and lunch. Each subject showed a significant reduction in the AUC of glycemia and insulinemia when the meals included the extract, compared to meals without it [16]. Therefore, the reduction in blood glucose levels observed with low-dose ABA in vivo in rats and humans occurs independently of increased insulin secretion. Another indication that the effect of ABA was independent of insulin is that the AUC values of glycemia and insulin measured in wild-type and TRPM2−/− mice, treated with ABA at 1 µg/kg BW along with an oral glucose load, were reduced in both genotypes [81], demonstrating that ABA stimulated muscle glucose uptake even in insulin-deficient animals [82].

In addition, in ex vivo experiments, skeletal muscle samples from fasted mice showed a twofold increase in the uptake of 18F-deoxyglucose (FDG) when exposed to nanomolar concentrations of ABA compared to untreated controls [81], confirming that the effect of ABA on muscle glucose uptake is insulin-independent. A comparable rise was observed in an in vivo experiment where rats received an oral glucose load (1 g/kg BW) without or with ABA (1 μg/kg BW). MicroPET imaging of the interscapular region showed that ABA enhanced FDG uptake by brown adipose tissue (BAT) in all the animals. This resulted in approximately a twofold rise in the average glucose consumption rate [83].

In a human study, consuming a daily food supplement with a vegetable extract providing around 1 µg ABA/kg BW led to significant improvements in metabolic rate over a period of 75 days. Key parameters used to evaluate the risk of metabolic syndrome and diabetes, such as fasting blood glucose, glycated hemoglobin, total cholesterol, LDL and HDL cholesterol, waist circumference, and body mass index, showed noticeable improvements. Additionally, the same study demonstrated that daily supplementation with an equivalent dose of synthetic ABA over four months enhanced glucose tolerance reduced glycated hemoglobin levels, lowered blood lipid concentrations, and led to weight loss in mice on a high-glucose diet [84].

### 2.3. ABA Stimulates Myocyte, Cardiomyocyte, and Adipocyte Glucose Uptake and Metabolism In Vitro, Ex Vivo, and In Vivo

#### 2.3.1. Skeletal Muscle

Skeletal muscle, which alone constitutes about 40% of total body weight in an average adult human, and body fat (white and brown adipose tissue, WAT and BAT, respectively), together represent the bulk of the cell mass in mammals [85,86,87]. Thus, a stimulatory effect of ABA on glucose uptake and metabolism by muscle and adipose cells would improve whole-body glucose management [88,89,90,91,92].

The first indication that nanomolar concentrations of ABA stimulated glucose uptake by rodent myocytes and adipocytes was obtained on L6 myoblasts and 3T3-L1 adipocytes, with a quantitative effect comparable to that of insulin, elicited by enhancing GLUT-4 translocation to the plasma membrane [24]. The absence of insulin in the in vitro experiments suggested that the effect of ABA was insulin-independent.

Subsequent experiments demonstrated that ABA stimulates glucose uptake by murine muscle biopsies ex vivo, an effect dependent upon the activation of AMPK [93,94,95,96,97,98], as it was inhibited by dorsomorphin [81,99]. Finally, the treatment of rats with 1 µg ABA/Kg BW significantly increased radioactive glucose uptake in skeletal muscle and BAT compared with untreated controls, resulting in a faster glucose clearance from the blood and a doubling of whole-body glucose uptake [83].

ABA-treated rats consequently showed a significantly reduced glycemia profile, with a lower plasma glucose area under the curve (AUC) compared with controls, while the plasma insulin AUC was approximately ten times lower, supporting the conclusion that the action of ABA reduces the amount of insulin secreted by β-cells in response to the peak of glycemia [16].

A further demonstration of the stimulation by ABA of muscle glucose uptake comes from studies on chronically ABA-treated mice, where skeletal muscle glycogen content increased and glucose tolerance improved. These mice also exhibit enhanced muscular performance when allowed free access to a running wheel: ABA-treated mice run approximately 2.5 times longer and cover a significantly greater distance (~2-fold) compared with controls, with a maximal speed 3.2 times higher than that of untreated mice [79]. In a separate study, an ABA-enriched extract at 0.125 µg/kg body weight improved glucose tolerance, insulin sensitivity, and fasting blood glucose in diet-induced obesity (DIO) and db/db mouse models, without increasing insulin levels. The extract also boosted muscle metabolism by enhancing the expression of genes related to glycogen synthesis, glucose and fatty acid metabolism, and mitochondrial function. However, the absence of LANCL2 in muscle tissue abrogated the effects of ABA, resulting in higher fasting blood glucose [100].

These effects of ABA on muscle cells depend upon the activation of AMPK, which in turn upregulates the expression of GLUT4 and GLUT1, mitochondrial respiration, and the expression of uncoupling proteins like sarcolipin and UCP3 [29,101,102,103,104]. As skeletal muscle is a major contributor to whole-body energy consumption, the stimulation of muscle glucose uptake and metabolism by ABA could significantly contribute to maintaining a normal body weight [105,106,107].

#### 2.3.2. Cardiomyocytes

Recent studies have explored the impact of the ABA-LANCL1/2 system on cardiomyocytes under both normal oxygen levels and hypoxic conditions. The results indicated that ABA and its receptors play a crucial role in mediating the cardiomyocytes’ response to hypoxia [39]. Under hypoxic conditions, the H9c2 rat cardiomyocyte cell line generates endogenous ABA, which subsequently activates the ABA/LANCL1/2 hormone/receptor pathway. The activation of this pathway triggers a signaling cascade that includes AMPK and PGC-1α, orchestrating a series of transcriptional and post-transcriptional events that promote the production of nitric oxide (NO). Interestingly, H9c2 cells with overexpression of LANCL1/2 and exposed to hypoxic conditions show a significantly enhanced mitochondrial proton gradient (∆Ψ) compared to cells where LANCL1/2 is silenced. These results suggest that the ABA-LANCL1/2 system may play a crucial role in maintaining mitochondrial ∆Ψ, thereby supporting metabolic energy production in cardiomyocytes during hypoxia and reoxygenation.

Mitochondrial function is crucial for maintaining the contractile and electrical activity of cardiomyocytes under normal conditions [108,109,110]. The ABA/LANCL1-2 system has been shown to enhance the mitochondrial number and ∆Ψ in normoxia, thereby boosting cellular respiration and energy production in cardiomyocytes [42]. This improvement enhances the cells’ structural and metabolic integrity, essentially promoting cardiomyocyte “fitness” through the AMPK/PGC-1α/ERRα signaling pathway and associated transcription factors. Additionally, the elevated basal and maximal respiration rates observed in LANCL1/2-overexpressing H9c2 cells, compared to silenced cells, are supported by an increased oxidative metabolic rate, particularly in glucose metabolism [42].

Indeed, in LANCL1/2-overexpressing cells, the transcription of glucose transporters GLUT4 and GLUT1, glycolytic enzymes (such as phosphofructokinase 1 (PFK1), glyceraldehyde-3-phosphate dehydrogenase (GAPDH), pyruvate kinase (PK), and pyruvate dehydrogenase subunit one (PDHα1)) increased approximately twofold compared to control cells. In line with the transcriptional increase, glucose uptake was approximately 40% higher in cells overexpressing LANCL1/2 compared to those with double silencing. Pre-incubation with 100 nM ABA further tripled glucose uptake in the overexpressing cells but had no effect on the silenced ones. Additionally, increased energy production in LANCL1/2-overexpressing H9c2 cells, as opposed to the double-silenced ones, was evident from a higher ATP/ADP ratio and elevated NAD^+^ levels. An elevated rate of respiration fueled by fatty acids was observed, accompanied by a twofold increase in the levels of proteins involved in fatty acid transport (such as carnitine palmitoyltransferase, CPT1β) and oxidation (such as acyl-coenzyme A dehydrogenase, ACADS). Additionally, the expression of fibroblast growth factor 21 (FGF21), a hormone that regulates energy metabolism at both the tissue and organismal levels, was also significantly increased in cells overexpressing LANCL1/2 compared to the control cells [42,111].

A recently discovered regulatory role of the ABA/LANCL1-2 hormone/receptor system has shown its involvement in protecting cardiomyocytes from ROS-induced oxidative stress [44]; the transcription factor ERRα is involved in this regulation [44,112,113]. In cardiomyocytes overexpressing LANCL1/2, a significantly higher survival rate was observed under oxidative stress conditions, along with reduced levels of lipid hydroperoxides and mitochondrial ROS. This protection stems from an enhanced enzyme expression profile that favors ROS-scavenging enzymes over ROS-producing ones. Additionally, other studies indicate that the protective effect against radicals provided by the activation of the LANCL1/2 receptor system is probably not restricted to a single cell type [114,115,116,117,118,119].

In summary, the overexpression of LANCL1/2 and their targeted stimulation via ABA treatment enhance the functionality and resilience of H9c2 cardiomyoblasts. The transcription factor ERRα orchestrates these significant transcriptional effects in LANCL1/2-overexpressing cells and interacts closely with LANCL proteins, forming a reciprocal feed-forward loop that strengthens transcriptional activation.

#### 2.3.3. Adipose Tissue

Adipose tissue (AT) is one of the largest endocrine organs in the body, playing a pivotal role in regulating energy balance, glucose, and lipid homeostasis, as well as impacting insulin sensitivity and satiety through the production of several adipokines. In mammals, white adipose tissue (WAT) primarily functions in energy storage, while brown adipose tissue (BAT) is responsible for energy expenditure for heat production. BAT is found in specific regions such as the cervical, supraclavicular, paravertebral, and supra-adrenal areas in adult humans [59]. Excess visceral WAT is strongly associated with metabolic syndrome, type 2 diabetes, and cardiovascular disease, primarily due to the low-grade systemic inflammation driven by adipokines produced in WAT and cytokines released by infiltrating inflammatory cells [120,121].

The finding that ABA can reduce WAT inflammation in mice on a high-fat diet is particularly significant, suggesting that ABA-containing nutraceuticals could be useful in mitigating low-grade inflammation in AT [122,123]. Indeed, in prediabetic individuals treated with an ABA-rich vegetal extract, high-sensitivity C-reactive protein (hs-CRP) levels, a marker of chronic inflammation, were markedly lower compared to untreated controls [56].

The activation of the LANCL1-2/PGC-1α/ERRα pathway in white and brown adipocytes, driven by LANCL protein overexpression and further augmented by ABA, leads to multiple metabolic and functional changes, which collectively enhance the energy-producing and energy-dissipating capacity of AT: (i) enhanced glucose transport through GLUT4 upregulation, mitochondrial biogenesis (mt-DNA), respiration (increased expression of complex I and greater basal and maximal oxygen consumption), and increased mitochondrial ∆Ψ; (ii) the upregulation of receptors for “browning” hormones (the β-adrenergic receptor ADRβ3, the thyroid hormone receptors THRα1/β); deiodinase, which converts T4 to the active T3; and uncoupling proteins UCP-1/3; (iii) a 2- to 4-fold increase in mitochondrial DNA and oxphos complex I (MT-ND1) in the BAT of ABA-treated mice; (iv) higher expression of browning hormone receptors in ABA-treated human adipocytes and in the BAT from ABA-treated mice; and (v) elevated expression of MT-ND1, thyroid hormone receptors and ERRα in LANCL1-overexpressing, LANCL2 KO mice compared to WT mice [57].

The ABA/LANCL System in BAT: In humans, BAT is much less abundant than WAT, even in lean individuals, and is primarily located around major arteries in the chest and abdomen. The mitochondria-rich brown adipocytes rely heavily on the PGC-1α/ERRα axis, which is crucial for their function. Both ERRα and ERRγ play a key role in controlling the expression of various genes involved in the differentiation of white and brown adipocytes [124,125]. In murine models, the complete loss of all three ERR transcription factors (α, β, and γ) in brown adipocytes led to a significant decrease in mitochondrial content, oxidative potential, and the ability to respond to β-adrenergic signals [126]. However, ERRα and ERRγ alone are capable of inducing this phenotype, while ERRβ is not required [127]. This functional redundancy between ERRα and ERRγ in brown adipocytes underscores ERRα’s critical role in regulating UCP1 transcription, a hallmark protein of BAT [128]. Research on mice with adipose tissue-specific deletion of folliculin (FLCN), a negative regulator of AMPK, showed that the constant activation of AMPK stimulates the PGC-1α/ERRα pathway in adipocytes [129]. This activation leads to reprogramming toward a brown adipocyte phenotype, characterized by increased mitochondrial mass, oxidative metabolism, UCP expression, energy expenditure, and thermogenesis, as well as resistance to high-fat diet-induced weight gain. In brown adipocytes, HDAC3, a histone deacetylase, serves as a crucial coactivator for PGC-1α and ERRα, driving the transcription of UCP1 [130,131]. BAT thermogenesis is physiologically activated by cold exposure and regulated through sympathetic innervation and thyroid hormones. Norepinephrine, released from sympathetic nerves, activates β-adrenergic receptors on BAT cells, triggering intracellular pathways that increase lipolysis and UCP1 expression. Additionally, thyroid hormones boost BAT’s thermogenic potential by promoting mitochondrial biogenesis and upregulating thermogenic genes [132]. Heat generation results from the partial oxphos uncoupling by means of proton transport across the inner mitochondrial membrane by “uncoupling” proteins (such as UCP1-3 and sarcolipin) and also through other proteins, such as the ATP/ADP transporter. These hormonal signals ensure that BAT remains responsive to changes in environmental and physiological conditions, such as cold exposure or diet shifts. By promoting oxidative metabolism and heat production, BAT increases energy expenditure, helping to combat weight gain and obesity. Understanding these regulatory mechanisms positions BAT as a potential therapeutic target for metabolic disorders, with a focus on increasing energy expenditure and improving metabolic health [133,134,135,136,137].

The ABA/LANCL hormone/receptor system has emerged as a key regulator of BAT activity due to its ability to target the AMPK/PGC-1α/ERRα signaling axis, triggering the transcriptional program controlled by these regulators, resulting in enhanced glucose and fatty acid oxidation and increased oxphos uncoupling (and thus heat production). This mechanism, known as non-shivering thermogenesis, leads to greater energy expenditure and enhanced fat oxidation, offering potential therapeutic benefits for managing obesity and metabolic disorders.

At both the transcriptional and protein levels, the ABA/LANCL system exerts direct transcriptional control over ERRα and PGC-1α in adipocytes, as the overexpression of LANCL1/2 leads to a 20-fold increase in ERRα mRNA levels, while the silencing of the LANCL proteins significantly decreases ERRα expression in both white and brown adipocytes. Additionally, ABA promotes ERRα expression in the BAT of WT mice, with spontaneous overexpression of ERRα observed in the BAT of LANCL1-overexpressing, LANCL2 KO mice [57].

By activating the AMPK/PGC-1α/ERRα axis, the ABA/LANCL system stimulates the expression of key respiratory chain complexes, enhances mitochondrial oxidative metabolism, and increases ATP production via a steeper mitochondrial ΔΨ. Additionally, ABA promotes the expression of UCP proteins facilitating the dissipation of the proton gradient across the inner mitochondrial membrane as heat. This uncoupling process is essential for thermogenesis, making ABA a promising therapeutic agent for enhancing mitochondrial function and reducing fat accumulation [12,57].

Moreover, ABA activates specific signaling pathways in adipocytes, influencing various metabolic functions. In adipose tissue, ABA primarily engages the mitogen-activated protein kinase (MAPK) pathway, particularly the p38 MAPK and extracellular signal-regulated kinase (ERK) pathways, to modulate metabolic processes. This activation enhances glucose uptake, increases lipolysis, and inhibits adipogenesis. ABA also interacts with the peroxisome proliferator-activated receptor gamma (PPARγ) signaling pathway, which is critical for regulating adipocyte differentiation and function. Through these pathways, ABA promotes oxidative metabolism and energy expenditure, contributing to improved metabolic health and reduced adiposity [93,138,139,140].

The ABA/LANCL System in WAT: ABA has recently been recognized as a modulator of metabolic processes in WAT. Unlike insulin, which promotes triglyceride (TG) synthesis and fat storage, ABA enhances glucose uptake and stimulates oxidative metabolism without inducing TG synthesis. This metabolic shift is mediated through the activation of the PPARγ pathway, which regulates adipocyte function and energy metabolism. Additionally, ABA influences inflammatory markers in WAT, potentially reducing the chronic low-grade inflammation typically associated with obesity and metabolic syndrome [81,141,142].

ABA has also been demonstrated to induce WAT browning by stimulating glucose uptake and enhancing oxidative metabolism. This process is characterized by increased mitochondrial activity and energy expenditure. Unlike insulin, which supports glucose uptake and lipid storage via TG synthesis, ABA primarily promotes catabolic pathways. This distinctive effect makes ABA a promising therapeutic target for addressing obesity and metabolic disorders. The signaling pathways activated by ABA and insulin in adipocytes differ significantly in their impact on energy metabolism. Insulin activates the PI3K/Akt pathway, which promotes glucose uptake, lipid synthesis, and fat storage through TG formation. In contrast, ABA activates the ERRα pathway, which is crucial for mitochondrial biogenesis and oxidative metabolism. The activation of ERRα by ABA enhances mitochondrial function, fatty acid oxidation, and energy expenditure, without promoting lipid storage. This difference in signaling pathways underscores ABA’s potential to improve metabolic health by enhancing oxidative metabolism and reducing excessive fat accumulation [12,57].

It has been suggested that activating the AMPK/PGC-1α/ERRα pathway pharmacologically could be a potential approach to promote “beige” characteristics in white adipocytes, which are more prevalent than brown adipocytes in humans. This approach aims to increase energy dissipation in WAT, thereby reducing body weight. As obesity and overweight rates continue to rise worldwide, both in developed and developing countries, research is increasingly focused on pharmacological and nutraceutical approaches to promote WAT browning. Animal studies on diabetes and fatty liver disease have yielded encouraging results when treated with natural bioactive compounds, showing enhanced energy expenditure and resulting in positive metabolic changes across the system [134,143,144,145,146].

The main effects of the ABA/LANCL2 system on myocytes, cardiomyocytes, and adipocytes are summarized in Figure 4.

## 3. Non-Overlapping Roles of ABA and Insulin

ABA and insulin may have evolved as two different signals in response to different conditions of glucose availability: when glucose is abundant, insulin activates metabolic pathways mainly responsible for the storage of metabolic energy into glycogen and triglycerides, for future use by the organism. Instead, under conditions of low/normal glucose availability, ABA activates metabolic pathways, resulting in ATP production, through the activation of mitochondrial oxidative metabolism [42,57], but also increases proton gradient dissipation and thus heat production [147]. Not surprisingly, the two very different metabolic programs triggered by insulin and ABA rely on different kinase master regulators: Akt for insulin and AMPK for ABA. Indeed, these kinases are linked by a mutual negative regulatory mechanism, whereby the activation of one results in the inhibition of the other [148]. What insulin and ABA do have in common is the activation of glucose transport, which, in the case of ABA, has been shown to be completely insulin-independent in muscle cells [79].

Thus, as summarized in Figure 5 we can identify the ABA/LANCL system as a new actor in the management of glycemia homeostasis and mitochondrial energy generation, as well as a potential new therapeutic target in pathological conditions caused, or aggravated by, mitochondrial malfunction, such as cardiovascular disease and diabetes mellitus [77].

## 4. Open Questions

### 4.1. Possible ABA Cell Sources

One of the open questions still awaiting an answer is related to the cell or tissue sources of plasma ABA. The role of the intestinal microbiota in producing ABA cannot be ruled out; some studies suggest that ABA might be produced by gut microbes, although this is still under investigation [149]. Recently, it has been demonstrated in vitro that different bacteria species of the human gut microbiota produce and release ABA in the presence of beta-carotene, an ABA precursor in vegetals, and that the oral intake of a formulation containing such bacteria and beta-carotene increases pABA in humans, up to a concentration sufficient to exert the hormone’s action on glycemic and lipidemic control [Magnone M, Zocchi E. “Composition for the treatment of dysbiosis of the intestinal microbiota” 28 December 2021; US 11,207,360 B2]. Bacterial cultures found in the animal gut, such as *Escherichia coli*, *Klebsiella pneumonia*, and *Proteus mirabilis*, have been shown to generate small amounts of ABA and other plant hormones [150]; however, there is a lack of studies investigating whether ABA or other plant hormones are present in the gut microbiome’s natural metabolome. Furthermore, it remains unclear whether the concentration and presence of these compounds are linked to various physiological conditions of the host [149].

These studies do not rule out the possibility that the gut microbiota contributes to pABA, although the fact that pABA increases rapidly (similarly to insulin) in humans after oral glucose load suggests that it is mostly produced and released from other cellular sources [24,75]. High glucose concentrations stimulate ABA release from rat insulinoma cells and human pancreatic islets [24,151]. In addition, GLP-1 stimulates the release of both insulin and ABA from the perfused rat pancreas, and vascular ABA in turn stimulates the release of GLP-1 from the perfused intestine [76], creating a possible feed-forward mechanism between GLP-1 and ABA secretion in response to high glucose. Human subcutaneous (white) adipose tissue was also found to release ABA in response to high glucose concentrations in vitro [24].

Hypoxia was shown to trigger the release of ABA from cultured rat cardiomyoblasts and the ABA/LANCL system in turn activated cardioprotective, eNOS-mediated NO generation [41]. ABA release by cardiomyocytes under hypoxia suggests exploring whether skeletal muscle may also produce ABA under stress conditions, such as intense physical activity. If this were the case, this finding would contribute to our understanding of the mechanisms underlying the beneficial effect of physical activity in improving glucose tolerance in diabetes. Intake of low-dose ABA was shown to improve glycemic control and reduce the dose of insulin required to reduce blood glucose in murine models of type 1 diabetes [78].

Finally, beta-pancreatic cells (human and murine) have been shown to release ABA when stimulated by high glucose levels [24,151]. This observation may be relevant to the understanding of the pathogenesis of hyperglycemia in T1D and to the observation that ABA improves insulin action in insulin-deficient mice: the demise of beta-cells in T1D would compromise the production of both glycemia-reducing hormones, insulin and ABA, only one of which is being provided by current therapeutic protocols.

### 4.2. Role of the ABA-LANCL System in Thermogenesis

The role of the ABA-LANCL1/2 hormone/receptor system in cell heat generation has recently been revealed from studies on human brown adipocytes and rat cardiomyocytes. In both cell types, the overexpression of LANCL1 and LANCL2 enhances, while their double silencing conversely reduces, the transcription and expression of mitochondrial OXPHOS uncoupling proteins (UCP1-3 in brown adipocytes, sarcolipin, UCP1, and ANT1 in cardiomyocytes). The logarithmic magnitude of the difference in the expression of uncoupling proteins in LANCL1/2-overexpressing vs. double-silenced cells is accompanied by an increase in mitochondrial DNA, respiration, and ATP generation in overexpressing vs. double-silenced cells [29,34]. As a result of these profound changes induced by overexpression or silencing of the ABA receptors on mitochondria, the cell’s power stations, one would expect overexpressing cells to generate more heat than double-silenced cells. Indeed, heat production by LANCL1/2-overexpressing H9c2 cardiomyocytes was found to be approximately double that produced by double-silenced cells, and the addition of ABA further doubled heat generation, as measured with two different methods (thermistor probes and differential scanning calorimetry). The difference in thermal output between overexpressing and double-silenced cells was abrogated by inhibitors of mitochondrial electron transfer, demonstrating the mitochondrial origin of the heat [72].

A possible role of the ABA-LANCL system in the regulation of thermogenesis is further suggested by the observation that the ABA/LANCL1/2 system controls the transcription of several proteins (hormonal receptors and enzymes) involved in thermogenesis, such as thyroid hormone and beta-adrenergic receptors and deiodinase (the enzyme converting the pro-hormone T4 into the active hormone T3) in human brown and white adipocytes [34].

Further studies are needed to establish the role of the ABA/LANCL system in thermogenesis at an organismic level, focusing on the tissues most relevant for mammalian body temperature control, i.e., brown and beige adipose tissue, skeletal muscle, and brain. Interestingly, these tissues are also those with the highest expression levels of the LANCL proteins in the human body (https://www.proteinatlas.org/search/LANCL+, accessed on 20 October 2024).

### 4.3. Is the ABA/LANCL System Linked to Genetic Abnormalities?

Genetic abnormalities associated with ABA and LANCL1/2 are relatively underexplored areas of research. However, emerging studies in these fields have uncovered potential links between these genetic variations and various physiological processes and diseases. LANCL1 expression was found to be increased at presymptomatic stages of amyotrophic lateral sclerosis (ALS), in the SOD1G93A transgenic mouse model, indicating a possibly significant role of LANC1 in this neurodegenerative disease [152]. In mice, four genetic regions on chromosome 1—Idd5.1, Idd5.2, Idd5.3, and Idd5.4—affect the risk of developing T1D [153]. These regions are connected to a change in the Ctla4, cytotoxic T-lymphocyte antigen 4, a gene that helps control immune responses by slowing down T-cell activity. LANCL1 is one of 11 genes in the Idd5.3 region that plays a key role in the strong protection from T1D. Together, the protective genes at Idd5 and Idd3 provide almost complete protection from diabetes, insulitis, and the formation of insulin autoantibodies [153]. Furthermore, the analysis of LANCL2 expression (data from Gene Expression Omnibus (*GEO*) repository ID:71047777) may potentially be associated with metabolic conditions such as type 2 diabetes or obesity. LANCL2 was significantly more expressed in the white adipose tissue (WAT) of insulin-sensitive individuals compared to insulin-resistant obese human subjects. Ongoing research into genetic pathways will offer a deeper understanding of the roles of ABA and LANCL1/2 in disease and their potential as therapeutic targets.

## 5. Future Perspectives

### 5.1. Nutraceutical ABA to Control Glycemia in Prediabetes and Diabetes: Preclinical and Clinical Studies

Several clinical studies, performed by independent groups, have demonstrated a significant improvement of the clinical chemistry parameters detecting glucose intolerance in subjects undergoing chronic ABA treatment. In a first, ground-breaking study, Magnone et al. showed that, at the same dose either administered as a pure molecule or present in fruit extracts, ABA improved glucose tolerance in rats and normal subjects, without increasing insulin release [16]. A follow-up study demonstrated that a dose of ABA (around 1 µg/Kg body weight) lowered hyperglycemia in mice fed a high-glucose diet. Additionally, this dose significantly reduced the area under the curve (AUC) for postprandial blood glucose, fasting blood glucose (FBG), and glycated hemoglobin (HbA1c) levels in healthy human participants [80]. These results were confirmed by a study employing a different ABA-rich fruit extract on healthy subjects [55].

Finally, recent studies exploring new sources of vegetal ABA to be used as nutraceuticals confirmed the beneficial effect of low-dose ABA on glucose tolerance in healthy humans [154] and extended this observation to prediabetic subjects [56].

The results of these studies on healthy and prediabetic subjects led us to the conclusion that ABA-rich vegetal extracts are both safe and efficacious in lowering FBG, HbA1c, and postprandial glycemia AUC and pave the way toward clinical studies on higher numbers of prediabetic and diabetic patients. These clinical results, along with those from in vitro studies on cell lines and in vivo and ex vivo experiments in rodents using nanomolar concentrations of ABA during treatment, are summarized in Table 2.

Indeed, there is ample room for improvement in antidiabetic therapy, both in T1D and in T2D. Nutraceuticals titrated in ABA could provide a cheap and effective addition to current and future therapies in the field of diabetes and prediabetes: in T1D, they would allow for a reduction in the dose of insulin required to achieve glycemic control, as suggested by recent in vivo results on insulin-deficient mice [78]; in T2D, they could contribute to improving glycemic control and, most importantly, provide treatment for prediabetes, a potentially reversible condition that still lacks a pharmacological treatment and is bound to increase exponentially worldwide in the near future.

In addition to the well-described effects on glucose tolerance, ABA, by targeting the LANCL/ERRα axis, exerts cardio- and neuroprotective effects, which are emerging from the most recent scientific literature and further increase its therapeutic potential in diabetes [51,53,155]. These effects are likely mediated at least in part by the stimulation of the antioxidant cell defenses induced by the activation of the LANCL/PGC1α/ERRα axis, as observed in cardiomyocytes [44] and possibly in other cell types as well.

An interesting aspect of ABA is its ability to inhibit the bitter taste receptor, a mechanism that affects both the gut microbiome and glucose metabolism. These bitter taste receptors are found in key areas where nutrient-derived ABA is present. This suggests that ABA produced by gut microbes could serve as a significant signaling molecule, facilitating communication between the microbiome and the host, especially given that microbes generate a variety of complex metabolites [156,157,158].

### 5.2. Does ABA Increase Body Temperature?

A very recent outcome of the ongoing research on the metabolic effects of targeting the LANCL1/2 receptor system with their ligand ABA or by overexpressing the proteins to stimulate glucose oxidation and mitochondrial respiration is the appreciation of their role in the mitochondrial generation of heat. By increasing the mitochondrial number, respiration, proton gradient, and oxphos uncoupling, the ABA/LANCL hormone/receptor system increases both mitochondrial ATP production and heat generation [57,147]. These effects should not be considered antithetical; indeed, mild uncoupling has been advocated to improve electron flow under conditions of limiting O_2_ availability, as occurs under hypoxia, or under conditions of accelerated energy metabolism, as triggered by the activation of the ABA/LANCL/ERRα axis [41,42]. The effect of increased heat production at a cellular level may translate into a higher whole-body temperature, similar to the effect of thyroid hormones. The interaction between the ABA/LANCL system and the thyroid hormone- and beta-adrenergic-mediated thermogenic effects on brown/beige adipocytes, skeletal myocytes and the central brain control of body temperature will be a fascinating new area of investigation.

## 6. Concluding Remarks

Several research groups have contributed to the present understanding of the role of ABA in glycemic control in humans, and the clinical use of ABA-containing nutraceuticals or food supplements to improve glucose tolerance in prediabetic and diabetic subjects can be foreseen in the near future. The fact that ABA-containing nutraceuticals can be produced at a relatively low cost from vegetable sources could also facilitate its commercialization in low- and middle-income developing countries, which will experience a projected logarithmic increase in prediabetes incidence in the coming decades [159], including the younger generations [160]. In addition to the implementation of healthy dietary and lifestyle strategies, which will require a very intensive effort toward the dissemination of information and education, ABA-containing food supplements could help to improve glucose tolerance and reduce the risk of an impending diabetes pandemic poised to dwarf the past COVID one.

However, much remains to be discovered regarding other important physiological functions of this pleiotropic hormone, potentially applicable to the clinical setting. The possible cardio- and neuroprotective properties of ABA could provide a valuable addition to its glycemia-controlling capacity in diabetic subjects and in the aging population in general. The possible role of ABA in enhancing whole-body energy expenditure and thermogenesis, suggested by in vitro studies, and to be confirmed by in vivo studies, makes it a possible adjuvant to combat obesity and a risk factor for the development of prediabetes and diabetes, as well as a condition predisposing to cardiovascular diseases, the first cause of death in the developed countries.

Hopefully, this review will help draw more scientists to the study of the manifold physiological functions of ABA, leading to the publication of new preclinical and clinical studies in the near future.

## Figures and Tables

**Figure 1 nutrients-17-00013-f001:**
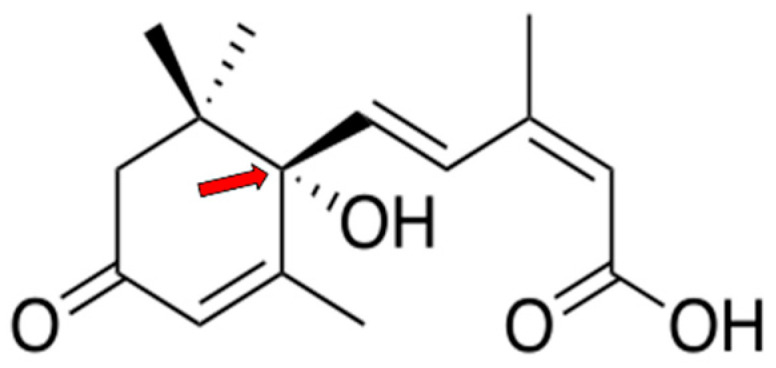
Chemical structure of 2-*cis*, 4-*trans* abscisic acid (ABA). ABA (MW 264) has a terpenoid structure and an asymmetric carbon, indicated by the arrow, giving rise to two enantiomers (+)- and (−)-ABA. Most functional activities in plants are attributed to (+)-ABA; in animals, both enantiomers appear to have biological activity [13].

**Figure 2 nutrients-17-00013-f002:**
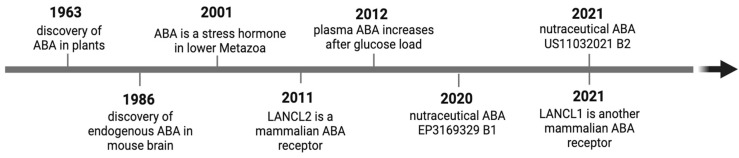
Milestones in the discovery of ABA as an animal hormone. More than 20 years elapsed since the discovery of ABA in plants to its first identification in the mammalian brain as an endogenous molecule. Subsequent studies on the role of ABA as a stress hormone in early Metazoa (marine sponges and hydroids) paved the way toward its identification as an endogenous mammalian hormone (2012) and the identification of its receptors, LANCL1 and LANCL2. In vivo studies on rodents and humans allowed for the filing and granting of patent applications in the EU (2020) and in the US (2021) regarding the use of ABA as a nutraceutical to improve glucose tolerance and metabolism.

**Figure 3 nutrients-17-00013-f003:**
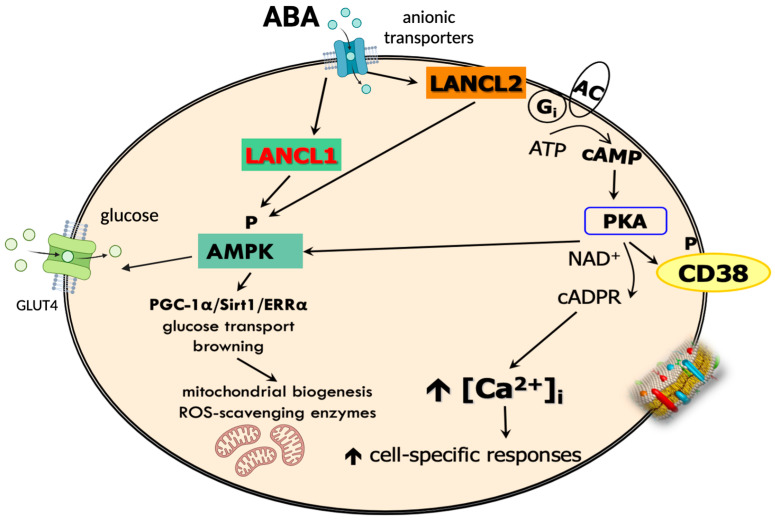
ABA-activated signaling pathways and primary functional effects of ABA on cellular metabolism. Deprotonated ABA, the predominant form at neutral pH, crosses the plasma membrane via an anion transporter [74] and binds to its receptors: cytosolic LANCL1 and membrane-anchored LANCL2. Both receptors can activate AMP-dependent kinase (AMPK) and the PGC-1a/sirtuin-1/ERRα axis, which in turn activates transcriptional programs controlling glucose uptake, mitochondrial function, and antioxidant defenses. Downstream of LANCL2, the activation of protein kinase A (PKA) also occurs, leading to phosphorylation and activation of the ADP-ribosyl cyclase CD38, with the production of cyclic ADP-ribose (cADPR) and ADP-ribose, both contributing to the generation of a cytosolic Ca^2+^ wave, via intracellular Ca^2+^ release from ryanodine-sensitive stores (cADPRs) and extracellular Ca^2+^ entry (ADPR).

**Figure 4 nutrients-17-00013-f004:**
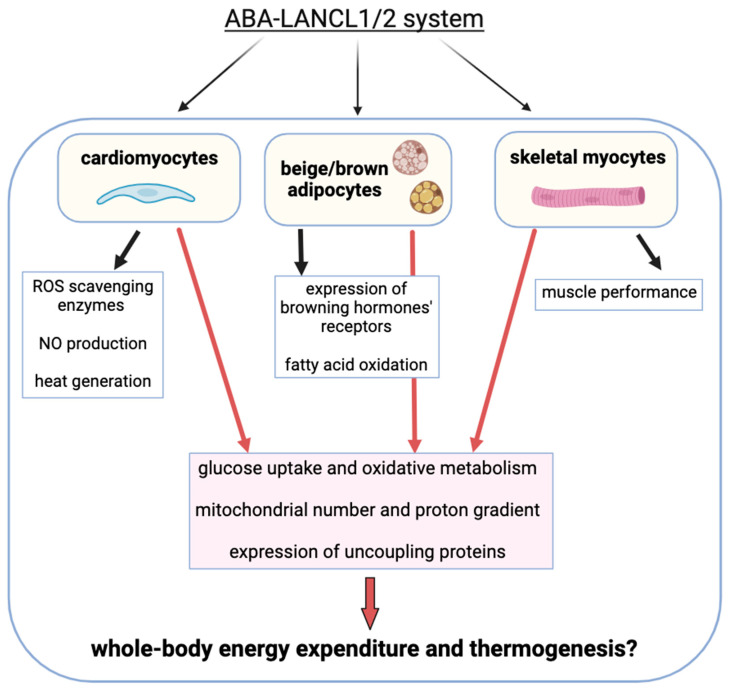
The ABA-LANCL1/2 receptor system enhances glucose uptake and mitochondrial oxidative function, boosts mitochondrial biogenesis and respiration, and elevates the expression of uncoupling proteins in cardiomyocytes, skeletal myocytes, and beige/brown adipocytes. In addition to these effects, the ABA-LANCL1/2 system also stimulates cell-specific functional features, such as NO generation and ROS-protection mechanisms in cardiomyocytes, the expression of thyroid and beta-adrenergic receptors in brown/beige adipocytes, and physical endurance. The increased heat generation observed in ABA-treated, LANCL1/2-overexpressing cardiomyocytes [124] suggests a possible control of thermogenesis by this hormone/receptor system also on muscle and beige/brown adipocytes.

**Figure 5 nutrients-17-00013-f005:**
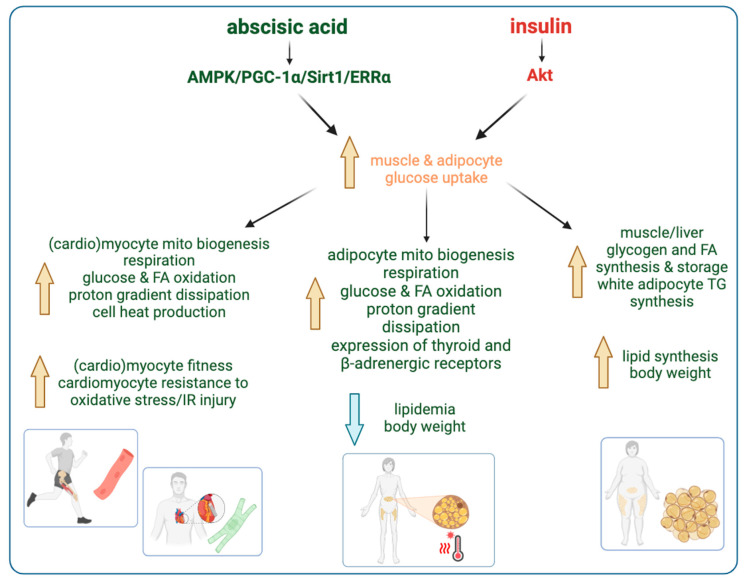
Non-overlapping functions of ABA and insulin in muscle and adipose tissue. Both ABA and insulin promote glucose transport in muscle and adipose tissues. Insulin increases the conversion of metabolic energy into storage forms such as muscle glycogen, fatty acids, and white adipocyte triglycerides, via the Akt pathway. Activated Akt suppresses AMPK. ABA, on the other hand, stimulates energy production by increasing mitochondrial mass and metabolic and respiratory activity. This in turn leads to increased heat generation, due to the concomitant activation of inner mitochondrial membrane proton leak, which reduces the ΔG for proton pumping and the retrograde electron flux, thus reducing ATP generation and leading to the production of oxidizing radicals [42,44,51].

**Table 1 nutrients-17-00013-t001:** Levels of ABA in different fruits and vegetables [11,16]. It should be noted that ABA content depends on the degree of fruit ripening and also on the specific cultivar.

Fruits	mg/Kg	Vegetables	mg/Kg
average content	0.62	average content	0.29
Avocado	2	Soybean	0.79
Citrus	1.25	Barley	0.20
Fig	0.72	Tomato	0.20
Bilberry	0.4	Wheat	0.15
Apricot	0.32	Pea	0.13
Banana	0.22	Cucumber	0.09

**Table 2 nutrients-17-00013-t002:** Potential ABA effects observed in in vitro and ex vivo experiments, as well as in rodents and humans following the administration of nanomolar ABA.

In Vitro Studies on Cell Lines	In Vivo and Ex Vivo Studies in Rodents	Clinical Studies in Healthy and in Prediabetic Subjects
ABA stimulates glucose consumption by adipocytes and myoblasts [24]	ABA improves glucose tolerance and reduces insulinemia in rats [16]; ABA increases blood glucose clearance and skeletal muscle uptake in rats [81]	An apricot extract providing a dose of ABA of 0.5 microg/Kg BW, taken before a carbohydrate-rich meal, reduces glycemia in healthy subjects [16]
ABA stimulates glucose uptake in the absence of insulin via an AMPK-dependent mechanism in rat muscle cells [81] and in murine muscle cells [29]	ABA enhances glucose tolerance and elevates muscle glycogen levels in TRPM2-KO mice with low insulin levels [81]	A 75-day administration of a vegetable extract providing a daily dose of ABA of 1.0 microg/Kg BW, reduces glycemia and lipidemia in borderline subjects [84]
ABA stimulates glucose uptake and metabolism-inducing brown features in rodent adipocytes [83]	An extract enriched with ABA enhances glucose tolerance and insulin sensitivity and reduces fasting blood glucose levels in mouse models of diet-induced obesity (DIO) and db/db mice [100]	Fig extracts enriched with ABA improve post-meal blood glucose and insulin levels in healthy adults [55]
ABA controls human adipocyte browning and energy expenditure via LANCL1/2 [57]	Chronically ABA-treated mice show increased skeletal muscle glycogen content, higher physical performance, and improved glucose tolerance [81]	A 3-month treatment with a dwarf peaches extract titrated in ABA improves glyco-metabolic and inflammatory parameters in prediabetic subjects [154]
Rat H9c2 cardiomyocytes overexpressing LANCL1/2 and treated with ABA display increased expression of glucose transporters GLUT4 and GLUT1, glycolytic enzymes, and pyruvate dehydrogenase, increased glucose uptake [42]	ABA improves glucose tolerance in LANCL2 KO mice by stimulating muscle GLUT4 expression via the LANCL1/AMPK/PGC1a axis [29]	

## Data Availability

The original contributions presented in the study are included in the article; further inquiries can be directed to the corresponding authors.

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
