# Peer review of "Role of Abscisic Acid in the Whole-Body Regulation of Glucose Uptake and Metabolism"

_nutrients, 2024, doi:10.3390/nu17010013_

Round 1
Reviewer 1 Report
Comments and Suggestions for Authors
The authors have provided a comprehensive review on the role of ascorbic acid in glucose absorption and metabolism. This manuscript is well-written and effectively discusses recent trends in ABS and its potential role in treating diabetes. However, before this manuscript can be published in Nutrients, minor revisions are required.
Comments:
-
Line 46: Ensure that cis and trans are italicized. Apply this formatting consistently throughout the manuscript.
-
Lines 50 and 64: Remove the hyphens at the beginning of the sub-titles. Ensure this correction is applied throughout the manuscript.
-
Line 119: Adjust the position of the subtitle for consistency with the formatting of other sections.
-
Title and Subtitles: Capitalize the first letter of each word in the title and all subtitles.
-
Figures and Schematics: The authors are advised to include more schematic representations illustrating the role of ABA in glycemia control. Specifically, consider depicting its effects on cardiomyocytes, skeletal muscles, and adipose tissues for better clarity and impact.
Author Response
Dear,
Herewith we are resubmitting a revised version of the review entitled “Role of Abscisic Acid in the Whole Body Regulation of Glucose Uptake and Metabolism”by Spinelli et al.
We would like to express our gratitude to the reviewer for their valuable suggestions. We believe the revised version of the review has been significantly improved, addressing all of the reviewers' comments. A detailed point-by-point response to the reviewers' feedback is included.
REVIEWER1
The authors have provided a comprehensive review on the role of ascorbic acid in glucose absorption and metabolism. This manuscript is well-written and effectively discusses recent trends in ABS and its potential role in treating diabetes. However, before this manuscript can be published in Nutrients, minor revisions are required.
Comments:
- Line 46: Ensure that cis and trans are italicized. Apply this formatting consistently throughout the manuscript.
Thank you for your comment; we have italicized "cis" and "trans".
- Lines 50 and 64: Remove the hyphens at the beginning of the sub-titles. Ensure this correction is applied throughout the manuscript.
We have removed the hyphens from the beginning of all sub-titles throughout the manuscript.
- Line 119: Adjust the position of the subtitle for consistency with the formatting of other sections.
Ok, we have adjusted the subtitle's position.
- Title and Subtitles: Capitalize the first letter of each word in the title and all subtitles.
Ok, we capitalized the first letter of each word in the title and all the subtitles.
- Figures and Schematics: The authors are advised to include more schematic representations illustrating the role of ABA in glycemia control. Specifically, consider depicting its effects on cardiomyocytes, skeletal muscles, and adipose tissues for better clarity and impact.
We included a new figure (Figure 4) that illustrates the effects of the ABA/LANCL2 system on skeletal myocytes, cardiomyocytes, and adipocytes.

Reviewer 2 Report
Comments and Suggestions for Authors
Please consider the following comments while you revise the content of this paper.
1- Please improve the quality of the illustrated figures.
2- There are several typographical and grammatical errors within the text. Please edit the whole text of the manuscript carefully.
3- The introduction section is too short. Please add some specific lines to this section and describe the topic of your paper in detail in this part of the manuscript.
4- Please provide a "search strategy" for this manuscript. Please determine which databases were screened to find relevant papers and how many were included in this review. Please clearly define the searched keywords and inclusion/exclusion criteria of searched documents.
5- Please convert some parts of the paper's text to the table.
6- Please add a conclusion remark to the end of your discussion.
7- Please add DOI identifiers to all cited papers.
8- Your paper has overlap with this paper that was published by the same authors: https://pmc.ncbi.nlm.nih.gov/articles/PMC7352484/pdf/nutrients-12-01724.pdf
9- Additionally, there were similarities between the content of this manuscript and another review paper published elsewhere: https://www.sciencedirect.com/science/article/pii/S2666351123000086
Please completely revise your manuscript and provide unique content to be published in this journal.
Comments on the Quality of English LanguageThere are several typographical and grammatical errors within the text. Please edit the whole text of the manuscript carefully.
Author Response
Dear,
Herewith we are resubmitting a revised version of the review entitled “Role of Abscisic Acid in the Whole Body Regulation of Glucose Uptake and Metabolism”by Spinelli et al.
We would like to express our gratitude to the reviewer for their valuable suggestions. We believe the revised version of the review has been significantly improved, addressing all of the reviewers' comments. A detailed point-by-point response to the reviewers' feedback is included.
REVIEWER 2
Please consider the following comments while you revise the content of this paper.
- Please improve the quality of the illustrated figures.
Thank you for your comment, we improved the quality of the figures
- There are several typographical and grammatical errors within the text. Please edit the whole text of the manuscript carefully.
We edited the text removing the typos and grammatical errors.
- The introduction section is too short. Please add some specific lines to this section and describe the topic of your paper in detail in this part of the manuscript.
Thank you for your valuable suggestion. We have incorporated specific lines into the introduction section based on your feedback.
- Please provide a "search strategy" for this manuscript. Please determine which databases were screened to find relevant papers and how many were included in this review. Please clearly define the searched keywords and inclusion/exclusion criteria of searched documents.
For this manuscript, we employed the NCBI PubMed database as our primary search strategy. We specifically used a combination of targeted keywords, such as “abscisic acid,” “mammalian,” and “human,” to ensure that the papers retrieved were focused on mammalian cells, rather than those related to plants. The search was extended to explore relevant literature on specific proteins and pathways, including LANCL1, LANCL2, LANCL3, ERRs (estrogen-related receptors), AMPK, UCP1, UCP3, sarcolipin, as well as white adipose tissue (WAT) and brown adipose tissue (BAT). To further refine the search and ensure the inclusion of comprehensive reviews, the term “review” was sometimes added. This strategy allowed us to gather a wide range of studies pertinent to our research, focusing on the molecular and cellular aspects of the relevant topics. In the updated version of the document, we have incorporated a total of 57 new references. These additions reflect the latest research and developments in the field, providing a more comprehensive and up-to-date overview of the topic.
5-Please convert some parts of the paper's text to the table.
We converted the preclinical and clinical results, along with those from in vitro studies on cell lines and in vivo and ex vivo experiments in rodents using nanomolar concentration of ABA during treatment, to the Table 2.
7- Please add DOI identifiers to all cited papers.
We added all DOI to all cited papers.
8- Your paper has overlap with this paper that was published by the same authors: https://pmc.ncbi.nlm.nih.gov/articles/PMC7352484/pdf/nutrients-12-01724.pdf
Thank you for your comment. The two reviews may be similar in the introductory section, where they provide general information on ABA and its historical discoveries. However, our review focuses on the most recent experimental research conducted after 2020. In particular, we have included all recent investigations conducted on cardiomyocytes, adipocytes, the new LANCL1 receptor, and the correlation between the ABA/LANCL system and the estrogen-related receptors ERRα. Figure 5 may suggest some similarities with the review cited, being the starting point of the same authors, but it includes all the latest news on the topic. This last scheme has however been slightly modified.
9- Additionally, there were similarities between the content of this manuscript and another review paper published elsewhere: https://www.sciencedirect.com/science/article/pii/S2666351123000086
Thank you for your comment. Honestly, we hadn't thoroughly reviewed the paper you suggested before your input, but we have now added it as a new reference. We believe the two reviews are distinct because ours is focused on glucose metabolism, while the “Biosensor” review provides a broader perspective on the role of ABA in various diseases such as Type 2 diabetes, malaria, Alzheimer's disease, glioma, and depression. Additionally, the “Biosensor” review emphasizes detection methods for ABA, a topic that is not covered in our review.

Reviewer 3 Report
Comments and Suggestions for Authors
Dear authors, your article is very interesting and presents some novelty; even so I recommend a few things:
Insert a table dividing in vitro, in vivo and clinical trials
Line 64: eliminate “-“
What is the daily recommended dose of this hormone? And what is the daily general consumption?
Line 119: rearrange this
Discuss the possible risks regarding its intake
Table 1: uniformize this table and insert a new line with the average content value
And the authors, what do you think? Do you think that ABA is better than insulin for treating diabetes? Do you think that it is better than insulin intake? And about other diseases?
Can its use exacerbate some comorbidities?
Comments on the Quality of English LanguageI recommend a strong revision regarding the language to clarify some sentences.
Author Response
Dear,
Herewith we are resubmitting a revised version of the review entitled “Role of Abscisic Acid in the Whole Body Regulation of Glucose Uptake and Metabolism”by Spinelli et al.
We would like to express our gratitude to the reviewer for their valuable suggestions. We believe the revised version of the review has been significantly improved, addressing all of the reviewers' comments. A detailed point-by-point response to the reviewers' feedback is included.
REVIEWER 3
Insert a table dividing in vitro, in vivo and clinical trials
We added the table
Line 64: eliminate “-“
We eliminated “-“.
What is the daily recommended dose of this hormone? And what is the daily general consumption?
The dose of abscisic acid that has been shown to produce the most favorable results in both in vivo animal studies and in healthy human subjects is 1 µg/Kg BW. This dosage has consistently demonstrated significant positive effects in different clinical studies (indicare refs), demonstrating its potential for therapeutic applications. Currently, purified, medicine-grade abscisic acid is not available, but commercial fruit extracts titrated in abscisic acid and providing approximately this dose are on the market.
The amount of ABA taken daily by a subject fed a typical Mediterranean diet, rich in fruits and vegetables, is approximately 10 times lower than that used in these studies.
Line 119: rearrange this
We modified the sentence.
Discuss the possible risks regarding its intake
A thorough investigation into possible toxic effects of ABA on animals is published at the EPA Federal Register / Vol. 75, No. 48 / Friday, March 12, 2010 / Rules and Regulations. The report concludes that abscisic acid cannot be allocated to any of the hazard categories for oral acute toxicity since the LD50 exceeds >5000 mg/Kg BW/day in rats; furthermore, the report concludes that ABA is in the same category IV (absence of toxicity) also for acute inhalation, skin irritation and dermal sensitization, that it does not have mutagenic effects in vitro and does not induce cell DNA or chromosomal damage in vitro. A 90-day chronic oral toxicity study concluded that the no observable adverse effect level (NOAEL) was 20,000 mg/kg/day. Finally, a prenatal developmental toxicity study (MRID No. 47470512) found no significant treatment-related reproductive effects or fetal abnormalities and established a NOAEL of 1,000 mg/kg/day.
On our part, we can add that ABA does not induce hypoglycemia in rodents at a dose 1000-times higher than that effective in lowering glycemia in both rodents and humans.
Table 1: uniformize this table and insert a new line with the average content value
We have made the required corrections.
And the authors, what do you think? Do you think that ABA is better than insulin for treating diabetes? Do you think that it is better than insulin intake? And about other diseases?
We do not suggest that ABA can replace insulin, when insulin is required due to the loss of beta cells. Indeed, we showed that in a murine model of streptozotocin-induced, insulin-dependent diabetes (T1D), ABA alone cannot substitute for insulin [78]. However, the same study also shows that ABA treatment can reduce the dose of insulin required to lower glycemia in diabetic mice. Thus, ABA in addition to insulin may be proposed in T1D to reduce the dose of insulin and its possible negative effects, i.e. increased lipid synthesis and the risk of hypoglycemia. In prediabetes and in overt T2D, ABA treatment may help to control glycemia by increasing glucose uptake in muscle and adipose tissue, thus reducing the stimulation by high glucose levels on beta-pancreatic cells to release insulin. Reducing the need for insulin secretion in turn could preserve beta cell function for longer time periods in a diabetic condition that typically evolves towards insulin deficiency due to eventual beta cell damage and loss. At variance with insulin, ABA stimulates metabolic energy consumption, not storage (a revised figure 5 summarizing this concept has been added to the manuscript). For these reasons, ABA treatment in T2D may represent a valuable adjunct to current pharmacological approaches, which include metformin (also an AMPK activator) and GLP-1 agonists (which induce further insulin release by beta cells).
Additionally, numerous studies conducted in animal models have shown the beneficial effects of ABA in treating a wide range of conditions, such as neurological and psychiatric disorders, cancer, and malaria infections [4, 5, 30]. These conditions often involve significant disruptions in inflammatory processes that contribute to their onset and progression. The anti-inflammatory properties of ABA have been shown to modulate the immune response and restore balance, helping to alleviate chronic inflammation that exacerbates these diseases. By targeting key inflammatory pathways, ABA appears to offer a promising therapeutic approach not only for symptom management but also for potentially preventing disease progression. This growing body of evidence highlights ABA’s potential in managing conditions with underlying inflammatory dysregulation, making it a promising avenue for novel therapeutic interventions.
Can its use exacerbate some comorbidities?
An answer to this question may come from future studies, arising from a more widespread perception of the possible benefits of ABA dietary integration. From what we know now, from studies on prediabetic subjects, inflammatory parameters such as hsCRP were significantly reduced in chronically ABA-treated subjects and this observation holds promise regarding a possible effect of ABA on the reduction of “low-grade inflammation”, currently believed to lie at the heart of several pathological conditions, including diabetes, cardiovascular diseases, autoimmunity and inflammatory bowel diseases. Indeed, in a murine model of IBD, ABA treatment ameliorated disease activity, colitis and reduced colonic leukocyte infiltration and inflammation [32]. From these studies, it could be expected that ABA treatment should reduce one of the principal pathogenetic mechanisms responsible for several chronic diseases.

Round 2
Reviewer 2 Report
Comments and Suggestions for Authors
Accept
Author Response
Thank you!!